# Functional Near-Infrared Spectrometry as a Useful Diagnostic Tool for Understanding the Visual System: A Review

**DOI:** 10.3390/jcm13010282

**Published:** 2024-01-04

**Authors:** Kelly Acuña, Rishav Sapahia, Irene Newman Jiménez, Michael Antonietti, Ignacio Anzola, Marvin Cruz, Michael T. García, Varun Krishnan, Lynn A. Leveille, Miklós D. Resch, Anat Galor, Ranya Habash, Delia Cabrera DeBuc

**Affiliations:** 1School of Medicine, Georgetown University, Washington, DC 20007, USA; ka798@georgetown.edu; 2Department of Ophthalmology, Bascom Palmer Eye Institute, University of Miami, Miami, FL 33136, USA; rxs1576@miami.edu (R.S.); mxa1441@med.miami.edu (M.A.); mtg84@med.miami.edu (M.T.G.); vxk156@med.miami.edu (V.K.); lal2014@med.miami.edu (L.A.L.); agalor@med.miami.edu (A.G.);; 3Department of Cognitive Science, Faculty of Arts & Science, McGill University, Montreal, QC H4A 3J1, Canada; irene.newman@mail.mcgill.ca; 4Department of Ophthalmology, Semmelweis University, 1085 Budapest, Hungary; miklosresch@gmail.com

**Keywords:** fNIRS, eye, virtual reality, artificial intelligence, augmented reality, visual systems, brain–computer interfaces, optical imaging, visual system, neuroimaging

## Abstract

This comprehensive review explores the role of Functional Near-Infrared Spectroscopy (fNIRS) in advancing our understanding of the visual system. Beginning with an introduction to fNIRS, we delve into its historical development, highlighting how this technology has evolved over time. The core of the review critically examines the advantages and disadvantages of fNIRS, offering a balanced view of its capabilities and limitations in research and clinical settings. We extend our discussion to the diverse applications of fNIRS beyond its traditional use, emphasizing its versatility across various fields. In the context of the visual system, this review provides an in-depth analysis of how fNIRS contributes to our understanding of eye function, including eye diseases. We discuss the intricacies of the visual cortex, how it responds to visual stimuli and the implications of these findings in both health and disease. A unique aspect of this review is the exploration of the intersection between fNIRS, virtual reality (VR), augmented reality (AR) and artificial intelligence (AI). We discuss how these cutting-edge technologies are synergizing with fNIRS to open new frontiers in visual system research. The review concludes with a forward-looking perspective, envisioning the future of fNIRS in a rapidly evolving technological landscape and its potential to revolutionize our approach to studying and understanding the visual system.

## 1. Introduction

The investigation of brain function and its intricate relationship with visual perception has long captivated researchers in neuroscience. Transmitting visual information from the eye to the brain is a complex sequence of events involving multiple anatomical structures and physiological events. The information from light reception and phototransduction is relayed through retinal circuits, optic nerves, the optic chiasm and thalamic nuclei. The brain processes these signals to create our conscious visual perception [1]. This intricate network of anatomical structures and neural processing stages ensures the interpretation and perception of visual stimuli within the brain.

Advancements in optical and neurological imaging techniques, particularly those applied in human brain research, have significantly contributed to unraveling the mysteries of brain processing. Techniques such as functional magnetic resonance imaging (fMRI), magnetoencephalography (MEG) and electroencephalography (EEG) have significantly contributed to understanding human brain activity patterns during visual tasks and stimuli presentation [2,3,4]. However, these techniques often have limitations, including restricted portability and susceptibility to motion artifacts, making them less suitable for specific populations and settings [5]. Among these techniques, Functional Near-Infrared Spectroscopy (fNIRS) has emerged as a promising tool for understanding the human brain’s complex workings due to its ability to measure changes in oxygenated and deoxygenated hemoglobin levels, thus providing insights into neural activity and functional connectivity [6].

fNIRS technology offers a novel approach to studying brain function, especially visual processing and perception. fNIRS offers unique advantages, such as portability, cost-effectiveness and safety, making it suitable for clinical and research applications. Additionally, the combination of fNIRS with emerging technologies like virtual reality (VR), augmented reality (AR) and artificial intelligence (AI) opens new avenues for immersive investigations into brain function [6].

As the field of neuroimaging continues to evolve, fNIRS stands out as a powerful tool for understanding the intricacies of the visual system. Its ability to capture hemodynamic responses and reflect underlying neuronal activity makes it a valuable addition to the neuroscientist’s toolkit. This review delves into the wealth of literature on fNIRS applications related to the visual system, particularly the mechanism of this technology, previous usage to understand aspects of the optical system and its potential to reshape the diagnostic landscape in clinical and research contexts.

## 2. Data Sources and Search Strategy

In this review, we conducted a systematic literature review of all published original research involving fNIRS in visual system studies to evaluate how fNIRS has been employed for investigating the visual system. The literature search, conducted in April 2023, began with an initial search of the PubMed, Scopus and Web of Science databases using the following search terms: Functional Near-Infrared Spectroscopy, fNIRS, eye function, eye disease, visual cortex, visual stimuli, visual system, virtual reality, augmented reality and artificial intelligence. These keywords were selected by consensus among the researchers. Additional spot-check searches from key references in identified works were also conducted to strengthen the reliability of the literature review further. Only English-language articles published in peer-reviewed journals were included. All studies involving fNIRS associated with research and clinical applications in humans extracted from the literature using the above keywords were included. Our literature search identified 31 unique papers in this area, which were included for further analysis.

## 3. Considerations of fNIRS

### 3.1. Mechanism of Action

Functional Near-Infrared Spectroscopy (fNIRS) is a wearable optical spectroscopy system initially developed for monitoring continuous and non-invasive brain function by measuring oxygenated and deoxygenated hemoglobin concentrations during neuronal hemodynamic responses [7]. Brain activity can be inferred from these hemodynamic fluctuations, as they are related to underlying neural activity through the neurovascular coupling phenomenon [8]. fNIRS measurements are captured by transmitting near-infrared light onto the scalp, attenuated by absorption and scattering [5]. This near-infrared light can penetrate biological tissues and be absorbed by chromophores, such as oxyhemoglobin (HbO) and deoxyhemoglobin (HbR) [9]. Oxygenated hemoglobin allows the monitoring of the hemodynamic response induced by different stimuli, as the amount of oxygen that reaches the activated brain region is higher than the rate at which it is consumed, generally increasing the amount of HbO and decreasing the amount of HbR (Figure 1) during cortical activation [5].

### 3.2. Historical Development

fNIRS technology has undergone significant advancements, allowing it to provide measurements of cortical activation. Frans Jöbsis first reported using near-infrared spectrometry to detect hemoglobin in 1977, with the original equipment being only a low-spatial-resolution single-channel continuous-wave system [11]. In 1984, the work of David Delphy expanded fNIRS functionally by creating a system that measured changes in oxygenated and deoxygenated hemoglobin [11] (Figure 2). Hamamatsu Photonics developed the first commercial system of fNIRS. In 1994, Hoshi and Tamura continued their progression by producing a 10-channel continuous-wave system. By the early 2010s, a wearable battery-operated single-channel system was created. Today, there are three main types of fNIRS technologies, including time-domain (TD), frequency-domain (FD) and continuous-wave (CW) technologies. fNIRS neuroimaging is progressively advancing in technical sophistication and gaining traction within neuroscience. For example, Figure 3 shows an innovative, portable, whole-head TD-fNIRS device from Kernel Inc. (Los Angeles, CA, USA) [12], a neurotechnology company revolutionizing precision neuroscience. This scalable technology captures high-quality cerebrovascular signals, provides whole-head coverage and is thus well-suited to study cortical connectivity patterns [13,14]. 

### 3.3. Advantages and Disadvantages

One of the most widely mentioned advantages of fNIRS compared to other neuroimaging techniques is its portability [5,9,14]. The physically small technology allows it to be moved easily around a clinical setting and brought to differing populations, as they may have physical or cognitive constraints limiting their ability to use other neuroimaging techniques [5]. fNIRS is additionally cost-effective and resistant to motion [11]. These qualities are highly beneficial for studying otherwise difficult populations, and the relatively low cost of fNIRS can provide researchers with differing funding opportunities to capture neuroimaging measurements. fNIRS also does not involve exposure to high magnetic fields and ionizing radiation, decreasing the risks involved with its utilization [8].

fNIRS has mixed reviews regarding its spatial and temporal accuracy, as some works claim it lacks accuracy and precision, whereas others state that its ability to provide these two qualities simultaneously is beneficial [2]. fNIRS does, however, have shallow photon penetration in individuals with dense skulls or thick, dark hair [3]. Individuals with these qualities tend to provide less accurate fNIRS measurements, if any, which is a main disadvantage to fNIRS [3]. However, its portability, cost effectiveness, resistance to motion, long-term monitoring and safety in operation provide beneficial qualities that show promise for its application in research and clinical settings.

### 3.4. Application in Other Fields

fNIRS has been used in other fields of study, including psychiatry, psychology and neurology. As pointed out by Piniti et al., fNIRS’s lightweight and quiet features have allowed for its usage in studying populations in which other neuroimaging techniques encounter difficulty [5]. fNIRS has shown the potential to study individuals with autism spectrum disorder (ASD), attention deficit hyperactivity disorder (ADHD) and schizophrenia who may otherwise have issues with the requirements of other imaging techniques [5]. Other works have explored pain using fNIRS, finding the technology useful for objective pain assessment in clinical environments [14]. Additionally, fNIRS has been used to evaluate cerebral oxygenation and autoregulation in patients with stroke and traumatic brain injury, providing a basis for this technology’s utilization in neurology [9].

## 4. Application to the Visual System

### 4.1. Understanding the Eye

Recent work has begun understanding the eye using fNIRS technology. In an early design study of NIRS technology, oxygenated and deoxygenated hemoglobin results of saccadic eye movements were compared against fMRI measurements of the visual cortex [16]. These preliminary findings introduced the NIRS design as a useful monitoring tool when supplemented with fMRI in patients with visual disorders [16]. Another study found that patients with an upward gaze and strong trigeminal proprioceptive evocation seemed to induce rapid oxygen consumption in the ventromedial prefrontal cortex [17]. This upward gaze activates sweat glands, causing deoxyhemoglobin production that regulates this physiological arousal [17].

fNIRS has additionally provided assessment measures for treating amblyopia [18]. In a study of 10 subjects, Iwata et al. found that, when patients were treated with both eyes open compared to one eye occluded, there was significantly greater activation of the visual cortex observed through oxygenated hemoglobin levels [18]. fNIRS also illustrates the potential to study the primary visual cortex of migraine patients, as Yamakawa et al. found fNIRS measurements that showed statistical significance when visual stimulation intensity was applied to intrinsically photosensitive retinal ganglion cells projecting to the primary visual cortex [19].

A recent study compared the degree of change in oxygenated hemoglobin caused by visual stimulation in emmetropic and myopic participants [20]. When comparing emmetropic and uncorrected myopic participants, the degree of change was higher in emmetropic participants. A higher degree of change in oxygenated hemoglobin was observed in corrected myopic patients than in uncorrected myopic patients. Myopia-induced emmetropic participants had a decline in functional activity; however, this was recovered when the myopia lens was removed. These results indicate that myopic defocus tends to reduce the level of oxygenated hemoglobin and functional activity in the visual cortex. However, visual cortical function can be restored with optical correction, as measured by fNIRS [20]. This literature illustrates the usefulness of fNIRS in establishing a further understanding of the eye generally.

### 4.2. Eye Function

fNIRS has additionally been used to comprehend eye function. Wijeakumar et al. utilized fNIRS technology to quantify the extent of occipito-parietal activation when high-contrast checkerboard visual stimuli were presented [2]. This study found that dynamic pattern presentation produced a more significant hemodynamic response correlated to the activation of V1 neurons compared to static checkerboard stimuli [2]. This difference in the response indicates the recruitment of many temporal frequency-selective neurons and neurons sensitive to spatial frequencies and contrast [2]. This functional usage of fNIRS provides a basis for using this technology to assess eye function.

Additional work has confirmed the reliability of fNIRS for functional mapping by verifying the vertical asymmetry of the visual cortex using fNIRS measurements [3]. By presenting black and white wedge checkerboard stimuli to the four visual field quadrants, oxygenated hemoglobin concentrations were consistently higher in the contralateral hemisphere of the simulation, reflecting the retinotopic organization of the visual cortex [3]. These results from fNIRS measurements corroborate with findings from previous neuroimaging techniques such as fMRI or MEG, confirming the reliability of fNIRS for retinotopic mapping and investigations of vertical/horizontal functional asymmetries within the visual cortex [3].

When performing a simultaneous fNIRS–EEG study, fNIRS illustrated good sensitivity to low-level sensory processing in visual and auditory stimulation [4]. fNIRS results clearly distinguish between visual and auditory modalities, with high area specificity and stimulus selectivity [4]. Yarmouth et al. assessed the reliability of fNIRS to evaluate visual cortex activation within frontal eye fields during vergence eye movements using intraclass correlation coefficient (ICC) values [21]. Oxygenated hemoglobin (HbO) signals had higher ICC values compared to deoxygenated hemoglobin (HbR) signals, indicating increased reliability for only oxygenated hemoglobin signals [21]. This result may be due to a noisier HbR signal resulting from a visible cardiac wave in the raw data signal acquisition, as well as the fNIRS instrumentation manufacturer having stated that HbR has a lower signal-to-noise ratio compared to that of HbO [21]. These studies demonstrate the capability of fNIRS for comprehending eye function.

### 4.3. Eye Disease

Other literature has also examined the utility of fNIRS for various eye diseases. One study used fNIRS to delve into cortical neural correlates of visual fatigue during binocular depth perception for different disparities [22]. fNIRS measurements revealed an increase in oxygenated hemoglobin in the occipital cortex during binocular depth perception, especially in particular regions of V1 with overlap in V1 and V2, confirming that the parieto-occipital cortices are spatially correlated with binocular depth perception [22]. The study also revealed that, once stereovision is established, hemodynamic responses approach a peak with an amplitude correlated with and determined by stereopsis [22]. This trend indicates that visual fatigue may result from generating but not sustaining stereovision [22].

Treatment for visual vertigo patients often includes habituation exercises using optic flow to address symptoms of dizziness, disorientation or impaired balance [23]. Hoppes et al. investigated individuals with visual vertigo. They utilized fNIRS measurements to explore cerebral activation during optic flow to determine if visual fixation modulates brain activity. This study found greater cortical activation in the bilateral fronto-temporo-parietal lobes when the optic flow was viewed with fixation, providing preliminary evidence of the use of a fixation target during these habituation exercises [23].

Primary open-angle glaucoma (POAG) is a neurodegenerative disease associated with stress and quality of life [24]. One study investigated the effects of meditation on subjective quality of life and brain oxygenation in patients with POAG [24]. This short-term meditation course was associated with significant improvements in changes in HbO levels in the intervention prefrontal cortex, indicating meditation’s usefulness alongside standard treatment in patients with POAG [24]. Additionally, this study illustrates how fNIRS measurements can be used alongside previously established measurements to assess various eye diseases.

In recent work on glaucoma, another study investigated reductions in occipital hemodynamic responses following visual stimulation in healthy controls compared to glaucoma patients using time-domain fNIRS (TD-fNIRS) [25]. A statistical analysis of fNIRS measurements of 98 total participants showed lower amplitudes of oxygenated and deoxygenated hemoglobin concentrations in glaucomatous eyes compared to healthy control groups [25]. This result indicates multi-level degeneration of visual pathways in glaucomic individuals, demonstrating the usefulness of TD-fNIRS measurements as a supporting tool for evaluating glaucoma pathology [25]. This work illustrates how fNIRS may be used to evaluate the progression of various eye diseases and assess treatment for these conditions.

### 4.4. Visual Cortex

Various techniques combining fNIRS with other statistical models or brain imaging methods have illustrated the validity of fNIRS measurements in measuring visual cortex activity. In 2004, Schroeter et al. applied the general linear model, previously used to analyze fMRI data, to fNIRS optical imaging techniques, as these methods allow frequency domain analyses whose results can be used to generate a map with a network structure of various brain regions [11]. The general linear model results found that, when a visual stimulus was presented compared to no visual stimulation, the concentration of HbO increased, whereas the concentration of HbR decreased [11]. This study also found higher activity in the occipitotemporal region when participants were presented with rotating colored figures, which likely corresponded to the visual cortex region that processes motion, V5 [11]. Total and oxygenated hemoglobin increased in areas corresponding to V1–V3 when the checkerboard stimulus was presented [11]. These works confirm the validity of fNIRS measurements for visual cortex monitoring through gathered oxygenated and deoxygenated hemoglobin concentrations.

In combination with functional network analysis, fNIRS has been used to understand character recognition in the occipitotemporal cortex [26]. Hu et al. gathered data on behavioral performance, fNIRS measurements and functional brain connectivity [26]. The results revealed that most participants could more accurately identify a real Chinese character when compared to an artificially produced pseudo-Chinese character. This change in accuracy positively correlated to higher oxygenated hemoglobin concentrations in the bilateral occipital, temporal cortex and left fusiform gyrus. There was a much faster reaction time and accuracy for checkerboard stimuli compared to pseudo and real character cases, possibly due to the lack of involvement of orthographic, phonological or semantic processing present during character cases [26].

Additional work investigated how fNIRS and EEG combined can measure visual cortex function generally [8]. The results revealed stronger contralateral motor and visual cortex activations when comparing left and right contrast and frontal region activation during the Stroop task [8]. This paper confirmed that the hemodynamic responses measured through fNIRS significantly correlate with the electrical activity of an EEG across both motor and visual tasks, illustrating the validity of fNIRS measurements as a form of visual cortex monitoring [8]. These findings generally indicate the validity of fNIRS signals in visual cortex activation measurements, consistent with previous literature and expectations.

### 4.5. Visual Stimuli

Various studies have also found correlations between differing visual stimuli and hemodynamic responses to these stimuli. Wijeakumar et al. used fNIRS and visual evoked potential to investigate the correlation between neural and hemodynamic responses to stereoscopic stimuli [2]. This study observed increased oxygenated and total hemoglobin concentrations across the N1–P2 complex [2]. fNIRS, compared to visual evoked potentials (VEPs), measured slow vascular changes due to neural activation with several seconds of latency [2]. However, there was a correlation between VEP amplitudes and HbO concentration for complex stimuli [2]. In concurrence with previous literature, this study suggested that the N1–P2 complex could be a marker of stereopsis in V1 [2].

Another study used EEG and fNIRS measurements to examine whether lower visual cortex activation for visual processing could be due to more efficient visual sensory encoding in cochlear implant users compared to controls [27]. These results found more prominent decreased responses to repeated visual stimuli in cochlear implant users and enhanced visual adaptation and lower visual cortex activation in these participants [27]. These fNIRS data support the hypothesis that cochlear implant users more efficiently process visual stimuli than controls [27].

Repetition suppression is a well-characterized response to a recent experience found in the perceptual cortices of adult brains [28]. One study using fNIRS aimed to determine if infants exhibit similar neural responses to repetition as adults and found little evidence of auditory repetition suppression in the temporal cortex and no evidence of visual repetition suppression in the occipital lobe [28]. However, repeated visual stimuli elicited consistent responses in the occipital lobe, suggesting that asymmetry in repetition suppression may be due to visual repetition suppression’s later development [28].

Hoppes et al. used fNIRS to examine cerebral activation during the optic flow of individuals with and without visual vertigo [23]. The primary finding of this study was decreased activation in the bilateral middle frontal regions of visual vertigo participants during optic flow, which may represent a different form of control for these individuals over the usually reciprocal inhibitory visual–vestibular interaction present in individuals without visual vertigo [23]. These preliminary findings may allow for additional fNIRS application in monitoring changes in cerebral activation in individuals with complaints of dizziness and disorientation symptoms of visual vertigo [23].

More recent work investigated the influence of flickering frequency on the amplitude of changes in the concentration of HbO and HbR [29]. The findings demonstrate that the checkerboard flickering frequency significantly affects HbO concentration changes, with the highest amplitude of HbO concentration changes observed at a checkerboard flickering frequency of 8 Hz [29]. This study allows for a further understanding of concentrations of hemoglobin changes at differing flickering frequencies of visual stimuli [29]. Overall, the above studies illustrate the usefulness of fNIRS technology in understanding the cortical activation of various visual stimuli.

## 5. Future of fNIRS Technology

### 5.1. Virtual Reality

Though fNIRS has been applied previously to the visual system, this non-invasive imaging technique has been integrated with VR technologies. This integration has shown the potential to assist medical professionals in making clinical decisions and offers promising future applications [6]. fNIRS, combined with adaptive visuomotor tasks, has shown potential in enhancing neural activity within the brain’s sensorimotor regions and hand motor function [30]. Two possible functions have been proposed for using fNIRS in VR therapy. One involves monitoring and delivering enhanced feedback on cortical activation areas during treatment. Moreover, the other function involves incorporating fNIRS into a brain–computer interface paradigm for therapy [31].

Building on previous research that established a link between high-intensity intermittent aerobic exercise (HIE) and increased neural activity [32], a new study employed fNIRS and immersive VR. In cases where patients may have been unable to engage in physical activity due to a physical condition, a study conducted by Burin et al. utilized the illusion of immersive VR to increase their heart rates successfully [33]. In these VR spaces, patients subjectively reported feeling as though they were moving themselves when they had only moved their virtual avatars and not their physical bodies [33]. The study demonstrated that this virtual training positively affected the patient’s cognitive functioning and corresponding neurologic substrates, even without physical movement [33]. These findings indicate that a sense of ownership can modulate body and brain reactions regardless of physical movement. By utilizing the immersive VR platform and measuring neural function with fNIRS, sedentary patients with neurological disorders and motor impairment can benefit from this phenomenon in clinical treatment [33].

The increased use of VR in medical applications has been combined with fNIRS to monitor the effects of the VR system. According to the simulation hypothesis, the neural networks of an action–observation system located in the motor cortices of the brain are activated during overt motor execution and observation or imagery of the same motor action on platforms like VR [34]. A study by Holper et al. aimed to demonstrate the effectiveness of using a VR system in neurorehabilitation by assessing its impact on brain activation [35]. This work confirmed that fNIRS recording does not hinder the use of the VR environment, a crucial development for applying VR-fNIRS technology in neurorehabilitation. Nevertheless, two factors could limit this technology’s accuracy and potential usefulness. First, intersubject variability is evident at the group level and more so at the individual level [35]. This variability may arise from differences in motor imagery ability and anatomical and physiological variations among subjects and warrants further investigation. Second, the recording side (uni- or bilateral hemisphere) and hand side (left or right hand used during motor or imagery tasks) need to be taken into consideration to understand the combined effect of these factors better [35].

In a separate exploratory investigation, Seraglia et al. aimed to assess the feasibility of incorporating fNIRS brain imaging in a study of immersive VR experiences [36]. These researchers utilized a virtual line bisection task and identified cerebral activation in the parietal and occipital lobes [36]. This study underscored the advantages of fNIRS’s non-invasive nature, which allows patients to maintain their natural position during the procedure [36]. These results add to the growing evidence that combining fNIRS and VR is viable for examining conventional neuropsychological disorders.

### 5.2. Augmented Reality

In contrast to VR systems, augmented reality (AR) enriches the user’s surrounding environment by superimposing virtual objects onto the real world as overlays. By visually exploring the statistical distribution of fNIRS patients who underwent dimensionality reduction with principal component analysis, Galati et al. identified common and unique interaction patterns associated with different states of cognitive load [37]. Hemodynamic activity from fNIRS (HbO, HbR) exhibited a similar pattern, suggesting that the users’ neural activity was linked to their interactions [37]. This evidence supports the combination of AR and fNIRS, as it can be adapted to support and investigate a variety of complex decision-making and problem-solving tasks applicable to numerous real-world uses, including teaching and learning applications.

Researchers have utilized an fNIRS-based augmented reality brain–computer interface (BCI) in simulated real-time settings to assist medical professionals in identifying the location and timing of pain in patients. Hu et al. positioned fNIRS optodes over the patients’ prefrontal cortex and primary somatosensory area to observe cortical activity, and individuals with sensitive teeth underwent thermal stimulation [38]. This cortical activity was then overlaid onto a participant’s head in real time through an optical see-through head-mounted display (HoloLens) device worn by the clinician [38]. Another group demonstrated the effective integration of fNIRS-based BCIs and AR technology to tackle a six-class issue using only one mental task and an fNIRS channel [39] (Figure 4). This approach capitalizes on the potential of AR technology to enable smooth interaction with the real world, which warrants further investigation in future research.

McKendrick et al. conducted a study to compare the cognitive effects of using an AR display (Google Glass) versus a handheld smartphone display during an outdoor navigation task [40]. They employed the fNIRS system to measure prefrontal cortex activity and incorporated two additional tasks to evaluate differences in mental workload and situational awareness while navigating [40]. The researchers determined that using an AR wearable display resulted in the least workload during specific working memory tasks and showed enhanced situational awareness based on their assessment of prefrontal hemodynamics [40]. 

Finally, one study examined the prefrontal cortex (PFC) oxygenation response when twenty-two male subjects were asked to maintain equilibrium while semi-immersed in a VR environment on a virtual tilt board (VTB) balancing over a pilot at a ±35° angle [41]. It was demonstrated that there was a bilateral increase in the oxygenation response of the PFC, which differed depending on the task’s difficulty [41]. These results could be useful for diagnostic testing and functional neurorehabilitation given its adaptability in the elderly and patients with movement disorders [41]. The progress made in combining AR with fNIRS highlights the potential of this technology, but more research is needed to bridge the divide between its use in research settings and real-world applications.

### 5.3. Artificial Intelligence 

Along with the latest cutting-edge technologies like VR and AR, integrating machine learning (ML) and deep learning algorithm data with fNIRS offers unprecedented opportunities. Current breakthroughs in ML have been used to explore complex and voluminous fNIRS data. Tanveer et al. explored the use of convolutional neural networks (CNNs) to classify the drowsy and alert states of thirteen healthy subjects while driving a car simulator. The CNN architecture resulted in an average accuracy of 99.3% while detecting drowsy and alert states [42].

Although fNIRS has gained popularity for its portability, among other advantages, data collection remains challenging in certain populations or cases, such as infants. This difficulty complicates the task for machine learning algorithms, which require large datasets to generalize effectively. Here, generative adversarial networks have been utilized to solve the data scarcity problem. Wickramaratne et al. used a generative adversarial network to augment the data obtained for the classification of a tapping task, i.e., whether a subject’s task is a left finger tap, right finger tap or foot tap based on the fNIRS signal [43]. The classifier obtained an accuracy of 80% using actual data and 96% using augmented data [43].

Recently, the neuro-imaging field has experienced a paradigm shift due to breakthroughs in deep learning. Although these advances have primarily shown results in fMRI research rather than fNIRS, they signal exciting prospects for mapping brain activity across various neuro-imaging modalities. In a recent study, Takagi et al. successfully used latent diffusion models—initially developed by Rombach et al. [44]—to reconstruct high-resolution images of 512 × 512 pixels from human brain activity, all without the need for fine-tuning their deep generative networks [45] (Figure 5). Similarly, Perpetuini et al. developed a model for a generalizable retinotopy classification using acquired optical brain signals from the visual cortex during visual stimulation consisting of a rotating checkerboard wedge, flickering at 5 Hz [46]. Their results could encourage the use of optical fNIRS signals in real-time BCI applications.

Remarkably, the scope of reconstruction extends beyond images to include auditory stimuli. Utilizing non-linear decoding models applied to intracranial electroencephalography (iEEG) data obtained from 29 epilepsy patients, Bellier et al. successfully reconstructed Pink Floyd’s ‘Another Brick in the Wall’ from individuals who were passively listening to the track [47]. Similarly, Denk et al. accomplished music reconstruction using functional magnetic resonance imaging (fMRI) data [48]. Although advancements in image and sound reconstruction have not yet been fully realized in the realm of fNIRS—partially due to limitations in spatial resolution and issues of signal contamination—we anticipate that future hardware innovations will progressively enhance the capabilities of fNIRS. Such advancements could potentially enable stimulus reconstruction in the years to come.

In conclusion, the synergistic amalgamation of sophisticated data analysis techniques, such as machine learning algorithms, with fNIRS has inaugurated an expansive array of unattainable opportunities merely a few years prior.

## 6. Conclusions

The rise of fNIRS in neuroscience research and clinical applications has been a groundbreaking phenomenon, providing complex insights into the brain’s inner workings and paving the way for discoveries in diagnosis and building effective therapeutics.

This manuscript has comprehensively reviewed fNIRS, tracing its origin from a rudimentary experimental low-resolution system to a sophisticated, non-invasive imaging technique. Its unique advantages over its incumbents, such as portability and lower cost, makes fNIRS an invaluable asset in neuroimaging modalities.

Our paper dives deep into the use cases of fNIRS, exploring the visual system, namely the eye, eye functions, eye diseases, how it connects to the visual cortex and its response to visual stimuli. Such investigations have expanded our foundational knowledge and have practical applications in developing targeted therapies and interventions for visual impairments.

Looking ahead, the future of fNIRS with the integration of other emerging technologies like VR, AR and AI holds enormous potential to push neuroimaging capabilities even further. However, accepting and recognizing the current limitations of fNIRS systems and their susceptibility to motion artifacts and limited spatial resolution are paramount. But eventually, continued work is needed to mitigate these limitations. In conclusion, fNIRS is a powerful tool for enhancing our understanding of the visual system, with exciting and promising avenues for further research and applications. Its versatile and practical non-invasive nature places it as a technology that could redefine our approach to neuroimaging and, by extension, our understanding of the human brain.

## Figures and Tables

**Figure 1 jcm-13-00282-f001:**
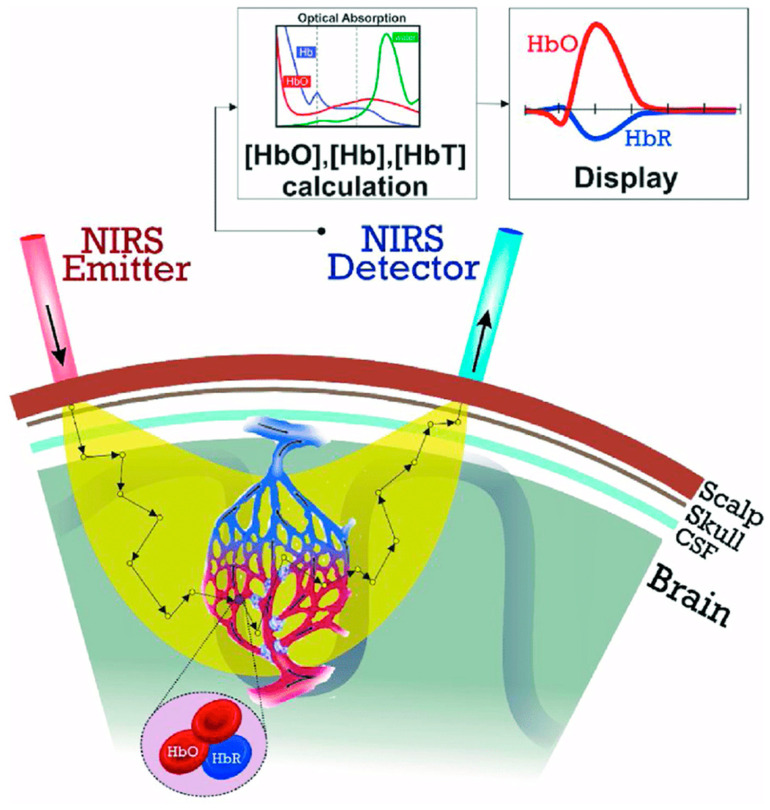
Schematic diagram of the fNIRS system. NIR light is generated and guided to the human’s head by optic fibers, and another fiber bundle diffusively reflects light from the head to detectors [10].

**Figure 2 jcm-13-00282-f002:**
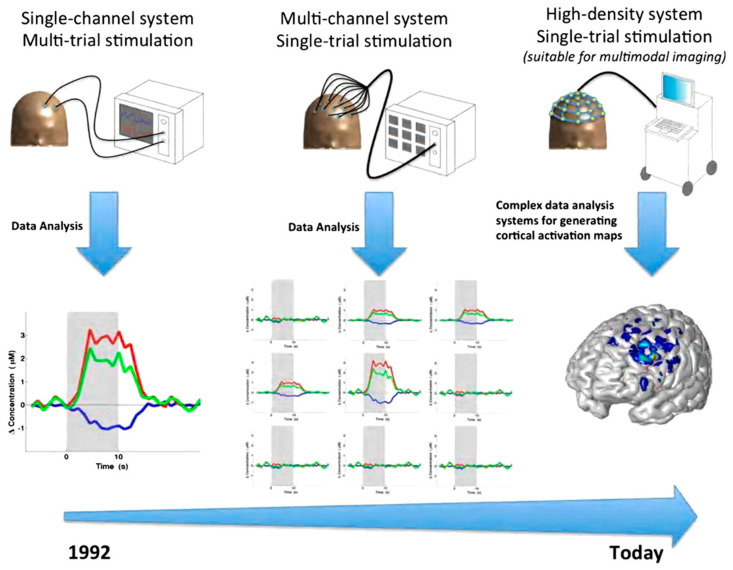
Sketch of the development of fNIRS instrumentation from 1992 (single channel with a low temporal resolution and poor sensitivity) up to multi-channel systems [13].

**Figure 3 jcm-13-00282-f003:**
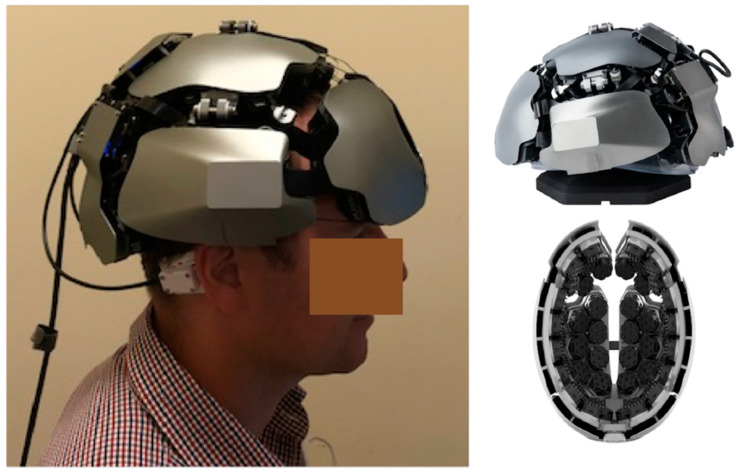
(**Left**): Side view of the Kernel Flow device (Kernel Inc. (Los Angeles, CA, USA)) for noninvasive optical brain imaging. (**Right**): Outer and inner views of the Kernel Flow headset [12,15] demonstrating the wearable form.

**Figure 4 jcm-13-00282-f004:**
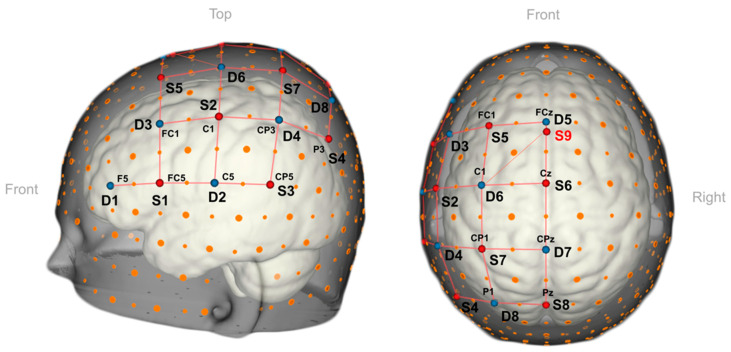
A 3D view of the fNIRS optode arrangement used in fNIRS-based BCIs integrated with AR technology [39]. The nine sources (S1–S9, red dots), eight detectors (D1–D8, blue dots) are placed over the left-hemispheric motor and premotor regions.

**Figure 5 jcm-13-00282-f005:**
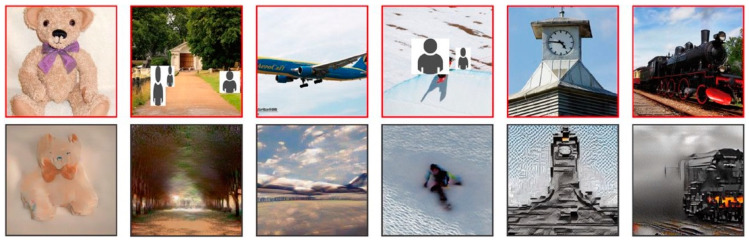
Images reconstructed from fMRI data from one subject: top row shows presented images, bottom row shows obtained images [45].

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
