# Peer review of "Functional Near-Infrared Spectrometry as a Useful Diagnostic Tool for Understanding the Visual System: A Review"

_jcm, 2024, doi:10.3390/jcm13010282_

Round 1

Reviewer 1 Report

Comments and Suggestions for Authors

The authors have done the literature review to show the ability of fNiRS for diagnostix purpose.

Althouh some literature is included but seems limited. The authors must use some systematic way of sorting the articles.

It is also not neccesary to use ony ref. that are related visual systems. It will be better to add the support of literature in terms of using fnirs as diagnostic tool such as studies on MCI and Parkinsons. such studeis must be included in the review.

The authors have used the figure of latest fnirs machine, which i am afraid not clinically tested, please double check for the ref.

The literature seems narrow. A little broad category needs to be inculcated. 

The authors are encourage to adhere the journals stye of refernces and inntext citation.

Author Response

Thank you for your valuable feedback and for the time you have invested in reviewing our manuscript.

Response to “The authors must use some systematic way of sorting the articles.”

To evaluate how fNIRS has been employed for investigating eye diseases/visual pathways, we conducted a systematic literature review of all published original research involving fNIRS in visual system studies. The literature search, conducted in April 2023, began with an initial search of the NCBI PubMed database using the following search terms: fNIRS, eye function, eye disease, visual cortex, visual stimuli, virtual reality, augmented reality, and artificial intelligence. Additional spot-check searches from key references in identified works were also conducted to strengthen the reliability of the literature review further. Only English-language articles published in peer-reviewed journals were included. All studies involving fNIRS in eye studies/visual pathways of any age were included. Our literature search identified 31 unique papers on eye studies/visual pathways topics, which were included for further analysis. We have added a new section to include the above (please see the "Data Sources and Search Strategy" section).

Response to “It is also not necessary to use only ref that are relation to visual systems comment.”

We appreciate your suggestion to include literature on using Functional Near-Infrared Spectroscopy (fNIRS) as a diagnostic tool in conditions such as Mild Cognitive Impairment (MCI) and Parkinson's disease.

We want to clarify that the primary focus of our review is to explore the role of fNIRS in advancing our understanding of the visual system. This focus was deliberately chosen to provide an in-depth and comprehensive analysis specific to this area. With its complex functions and significant research attention, the visual system presents unique challenges and opportunities for applying fNIRS. We aimed to specifically address these aspects in detail, contributing to a segment of the literature where we perceived a notable gap.

In the section elaborating on “applications in other fields,” we mentioned that fNIRS has been used in other fields of study, including psychiatry, psychology, and neurology as cited in Ref. 5, which is a comprehensive review that includes cognitive neuroscience area. Including broader applications of fNIRS, such as its use in diagnosing MCI and Parkinson's disease, while undoubtedly valuable in the general context of fNIRS research, would have expanded the scope of our review beyond its intended focus. Maintaining this focus enhances the depth and clarity of our analysis and offers valuable insights specific to researchers and practitioners interested in the visual system and its study through fNIRS.

We acknowledge the significance of the applications of fNIRS in diagnosing and understanding neurological conditions like MCI and Parkinson's disease. We agree that these areas of study are crucial and deserve thorough investigation. However, incorporating these topics into our current review would diverge from its targeted narrative and dilute the specialized focus on the visual system.

In conclusion, we appreciate your insightful suggestions and hope our rationale for our paper's specific focus is clear. Our review, with its current scope, provides a meaningful contribution to understanding the application of fNIRS in visual system research.

Response to “The authors have used the figure of latest fNIRS machine which I am afraid not clinically tests, please double check for ref.” 

We added a statement on lines 96-99 and the necessary citations.

Response to “The literature seems narrow.”

Please see our response regarding the systematic review methodology, which answers this concern above. 

Response to “The Authors are encouraged to adhere the Journals style of references and in text citations.”

Thank you for this recommendation. We have changed the format of the citations to the journals suggested ‘MDPI ACS Journals’ style on Endnote.

Reviewer 2 Report

Comments and Suggestions for Authors

Dear Authors,

I enjoyed your review on functional near-infrared spectrometry. I think it provides a thorough overview of the technique and its application in clinical neuroscience, including future uses in conjunction with emerging techniques. I only have a couple of observations:

1. In the introduction, in the paragraph that creates the context for fNIRS (lines 44-49) I think it has to be mentioned that only techniques used in humans are mentioned or introduce them in the larger context of brain research (like electrophysiology in all its applications, including intraop recordings in humans).

2. At lines 96-97 the same sentence is repeated ("Today, there are three main types..)

Author Response

Thank you for your valuable feedback and for the time you have invested in reviewing our manuscript.

Regarding the comment about the need to mention that the review included only techniques/applications used in humans, we have made it clear in the introduction as recommended. Please see paragraph/lines 41-51.  Also, the repeated sentence was eliminated. 

Round 2

Reviewer 1 Report

Comments and Suggestions for Authors

The authors have addressed the concerns.